# Research of Software Defect Prediction Model Based on Complex Network and Graph Neural Network

**DOI:** 10.3390/e24101373

**Published:** 2022-09-27

**Authors:** Mengtian Cui, Songlin Long, Yue Jiang, Xu Na

**Affiliations:** 1Key Laboratory of Computer System, State Ethnic Affairs Commission, Southwest Minzu University, Chengdu 610041, China; 2Swiss Center for Data and Network Sciences, University of Fribourg, 1700 Fribourg, Switzerland; 3Faculty of Business, Economics and Informatics, University of Zurich, Rämistrasse 71, 8006 Zurich, Switzerland

**Keywords:** software defect prediction, graph convolutional neural network, complex network, community detection

## Abstract

The goal of software defect prediction is to make predictions by mining the historical data using models. Current software defect prediction models mainly focus on the code features of software modules. However, they ignore the connection between software modules. This paper proposed a software defect prediction framework based on graph neural network from a complex network perspective. Firstly, we consider the software as a graph, where nodes represent the classes, and edges represent the dependencies between the classes. Then, we divide the graph into multiple subgraphs using the community detection algorithm. Thirdly, the representation vectors of the nodes are learned through the improved graph neural network model. Lastly, we use the representation vector of node to classify the software defects. The proposed model is tested on the PROMISE dataset, using two graph convolution methods, based on the spectral domain and spatial domain in the graph neural network. The investigation indicated that both convolution methods showed an improvement in various metrics, such as accuracy, F-measure, and MCC (Matthews correlation coefficient) by 86.6%, 85.8%, and 73.5%, and 87.5%, 85.9%, and 75.5%, respectively. The average improvement of various metrics was noted as 9.0%, 10.5%, and 17.5%, and 6.3%, 7.0%, and 12.1%, respectively, compared with the benchmark models.

## 1. Introduction

Software defect prediction is an indispensable part of software development because it can reduce the time and energy required for software testing during development. Software defect prediction is divided into two parts: the construction of software metrics [1], which is to count the features in the software code, and the model design, which is involved in the design of corresponding algorithms for different learning tasks and software metrics to achieve software defect prediction.

Traditional machine learning methods directly use software code features (such as changes in data and previous defects) to classify software defects. For example, Liu et al. [2] solved the cumulative unbalance problem using the SMOTE (synthetic minority oversampling technique) algorithm and solved the data noise problem using the ENN (extended nearest neighborhood) algorithm, as well as optimized the four-layer BP (backpropagation) network using the simulated annealing algorithm, and predicted the classification. Bashir et al. [3] proposed a feature selection method based on maximum likelihood logistic regression, which was beneficial to the selection of optimal feature subsets and can predict defect modules more accurately. Goyal [4] proposed a new filtering technique to effectively predict defects using support vector machines for the imbalanced data classification problem. The input of the prediction model based on machine learning is dependent on the software measurement elements; therefore, it needs to be changed continuously with the development of the software, which can potentially waste substantial time and energy in the reconstruction of the software measurement element.

With the development of deep learning, success has been achieved in NLP (natural language processing), image, audio, etc.; scholars used deep learning to learn deeper semantic features in code. Farid et al. [5] proposed a hybrid model to extract the semantics from an abstract syntax tree (AST) using a convolution neural network (CNN), and then used Bi-LSTM (bidirectional long short-term memory) to preserve key features while ignoring other features to improve the accuracy of software defect prediction. Deng et al. [6] felt that neural networks in NLP were more capable of learning semantic and contextual features in source code, firstly by extracting the code’s abstract syntax tree, which was then fed into the LSTM (long short-term memory) network, and then a prediction was made on where the file was defective or not. The above methods all took classes or files as research goals and did not consider the relationship between classes or files.

The software is mapped into a graph/network using the theory in the complex network, and the software defect prediction is carried out by studying the graph structure of the software. Šubelj and Bajec [7] found these existing community structures by mapping software into a dependent class-based network and proposed different applications of community detection in software engineering. Zhou et al. [8] used two measures of package cohesion and coupling, based on complex network theory, to verify the impact of code structure on software quality. Following the success of graph neural networks, Qu and Yin [9] mapped the software as a dependent class-based network, using different graph embedding techniques to embed the nodes of the graph into a d-dimensional vector space, the idea of embedding is to keep connected nodes close to each other in the vector space. The feature information can be learned from the graph structure of the software, but the above methods only consider the graph structure and ignore the node level features in the graph.

The software defect prediction model based on machine learning and deep learning treats the software module as a single unit, ignoring the interaction between software modules. The software defect prediction model based on the complex network only considers the graph structure of the software, ignoring the properties of the software module itself. Here, in this research, a software defect prediction model based on the complex network and graph neural network is presented. Firstly, the software system is mapped to a graph structure, with the classes as nodes, the dependencies between classes as edges, and traditional metrics as node attributes. Then, the whole graph is broken down into several subgraphs. Lastly, the information of the graph is learned through a multilayer graph neural network. Weights are given to each layer to prevent information loss.

The key contributions of this paper are as follows:(1)The application of the graph neural network in the complex network to make software defect prediction, followed by the use of the graph neural network to combine the structure of the software class graph along with the software’s class-level measurement element (node-level features, e.g., prior fault and new data) to learn new feature vectors. This represents an additional consideration in our model, compared with previous models, which only considered software graph structure or software defect measurement elements.(2)Use of the community detection algorithm to decompose the software graph structure into multiple subgraphs, and use of all the subgraphs as the input of the graph neural network model. This further simplifies the software graph structure, and the learned graph structure is a closely related subgraph.(3)Improvement of the graph convolutional neural network, such that the graph neural network can learn the graph structure features that are conducive to software defect prediction.

The remainder of this paper is organized as follows: Section 2 introduces the background knowledge of software diagram structure, then introduces community detection algorithms, and finally proposes a framework for software defect prediction. Section 3 presents the experimental environment, evaluation metrics, experimental setup, and experimental procedure. Section 4 discusses the results. Section 5 provides the conclusions and future work.

## 2. Materials and Methods

### 2.1. Software Diagram Structure

#### 2.1.1. Complex Network

The complex network [10] is a method for analyzing complex systems. Complex networks can abstract complex systems into graphs, and help people understand complex systems by analyzing some characteristics of graphs. Complex networks have been developed from the original Seven Bridges of Konigsberg problem [11] of network science. Telecommunication networks, computer networks, biological networks, cognitive semantic networks, social networks, etc., are all common complex networks in life, all of which are treated by different elements in the system as nodes, with connections between elements as edges.

#### 2.1.2. Software Class Depends on the Network

The software is a complex system; hence, it can be easily abstracted as a network for analysis. The classes in the software source code are regarded as nodes in the network, while the dependencies between classes are regarded as the edges of the network [12]. In software defect prediction, the node itself also has software defect measurement meta-information; thus, the node information is also regarded as a part of the software graph network.

### 2.2. Community Detection

By using the network’s structural information, community detection partitions the network into various smaller subnetworks. Nodes inside a community are closely connected, while nodes between communities are less connected. Depending on the type of network, community detection can be divided into two categories: static network community detection and dynamic network community detection [10]. The modularized community partitioning algorithm is a representation of the static network community partitioning technique. Modularity Q was first presented by Newman and Girvan [13] in 2004 to assess the effectiveness of community division. Numerous academics have devised analogous techniques by optimizing the Q-value in response to the modular Q suggestion. Among them, the Louvain algorithm [14] proposed by Blondel et al. is widely used because of its ability to quickly discover communities. The Louvain algorithm can be divided into two stages:(1)Every node starts off as a community. If a node’s modular gain from its current community to the community of its neighboring nodes is more than zero, the node will become affiliated with the community of its adjacent nodes, and its community affiliation will change. On the other hand, the initial community will be preserved until any node’s community change does not result in a modular gain that is more than zero.(2)A new network is created using the community acquired in the previous step as a node. The connection weight between nodes is the sum of all nodes in the original network between the two communities. The weight of the nodes, which have a self-circulation, is the total number of connections between the initial nodes in the community. When there is no gain update, step 1 is repeated for the new network.

### 2.3. Graph Neural Network

The processing object of the graph neural network is the graph which generally represents non-Euclidean relationships. The concept of a graph neural network (GNN) was proposed in 2005 [15]; later, in 2009, Dr. Scarselli [16] defined the theoretical basis of GNN. With the success of convolutional neural networks, scholars have thought about integrating the ideas of convolutional operators into GNNs, which are also known as graph convolutional neural networks (GCNs). There are two types of GCNs based on the spectral domain and spatial domain [17].
(1)GCNs based on the spectral domain include SCNN (spectral CNN) [18], ChebNet (Chebyshev spectral CNN) [19], and GCN [20]. The spectral domain convolution maps the graph topology into the spectral domain through discrete Fourier transformation, and then defines its graph convolution operator. The GCN convolution process can be represented by the following formula:(1)$$H^{\left(l+1\right)}=\sigma \left({\tilde{D}}^{-\frac{1}{2}}\tilde{A}{\tilde{D}}^{-\frac{1}{2}}H^{\left(l\right)}W^{\left(l\right)}\right),$$
where $\tilde{A}=A+I_{N}$ is the adjacency matrix of the undirected graph with added self-connections, $I_{N}$ is the identity matrix, ${\tilde{D}}_{ii}=\sum_{j}{\tilde{A}}_{ij}$, $W^{\left(l\right)}$ is a layer-specific trainable weight matrix, $H\in R^{N\times D}$ is the activation matrix of layer *L*, and $H^{\left(0\right)}=X$. $\sigma \left(\cdot \right)$ denotes an activation function.(2)GCNs based on the spatial domain include GraphSAGE (graph sample and aggregate) [21], GAT (graph attention network) [22], and GIN (graph isomorphism network) [23]. Spatial convolution aggregates the feature vectors of the first-order adjacent nodes of a node and then combines them with feature vectors of the current node. The graph convolution formula of GIN is as follows:(2)$$h_{v.}^{\left(k\right)}=MLP^{\left(k\right)}\left(\left(1+\varepsilon^{k}\right)\cdot h_{v}^{\left(k-1\right)}+\sum_{u\in N_{\left(v\right)}}h_{u}^{\left(k-1\right)}\right)$$

First, a graph G (V, E) is defined, in which v∈V, the feature vector of each node is $X_{v}$. $h_{v}^{\left(k\right)}$ denotes the representation vector of node *v* at the *k*-th layer, where *k* denotes the iteration level; $h_{v}^{\left(0\right)}=X_{v}$. $N_{\left(v\right)}$ represents a group of adjacent nodes of *v*. *MLP* is a multilayer perceptron. ε is a learnable parameter or a fixed parameter.

### 2.4. Software Defect Prediction Model Based on Complex Network and Graph Neural Network

This model primarily examines software from the perspective of the complex network, abstracts the software into a graph network, learns the representation vector of nodes using a graph neural network, and categorizes nodes on the basis of the representation vector. Figure 1 depicts the overall layout of the framework. Basically, it consists of two steps. Processing the data is the first phase, followed by using the class as the research granularity, abstracting the software source code into multiple nodes, and creating a network or graph using the dependencies among classes. Lastly, community detection techniques are used to split the graph into various subgraphs. The edge-link relationship is stored in the adjacency matrix in the second phase, and the adjacency matrix and the node-level features are seen as the structural information of the graph, which are considered as input to the graph neural network to obtain the representation vector of the node. Lastly, the multilayer perceptron (MLP) is used to classify the nodes. Section 2.4.1 analyzes the first part of Figure 1, and Section 2.4.2 analyzes the second step of Figure 1.

#### 2.4.1. Data Processing

The current software defect prediction models ignore the interdependence of the complex system in the software code. In order to map software systems into a graph, this research abstracts software systems from the perspective of complex networks [24]. In order to obtain the class dependency graph, we use a well-known technique. Additionally, the software defect measurement components of the class are taken into account as a node-level feature vector X, and the class dependence is transformed into an adjacency matrix A. Consequently, G can be used to represent the software graph (A, X). To further simplify the software graph and to make the learned representation of the graph more effective, we decided to use the Louvain algorithm to divide the graph into different subgraphs. Specific steps are as follows:
(1)First, a modularity *Q* is defined, which is used to judge the quality of the division; its value is between −1 and 1. The formula is as follows:(3)$$Q=\frac{1}{2m}\sum_{i,j}\left[A_{ij}-\frac{k_{i}k_{j}}{2m}\right]\delta \left(c_{i},c_{j}\right),$$
where *m* is the number of network connections, and *i*, *j* represent any two nodes in the network. When they are connected, $A_{ij}$ is 1; otherwise, it is 0. $k_{i}$ indicates the degree of node *i*. $c_{i}$ indicates the community of node I, and $\delta \left(c_{i},c_{j}\right)$ is used to judge whether nodes *i* and *j* are in the same community. If so, it is 1; otherwise, it is 0.(2)Initially, each node belongs to a community, and there are several communities with several nodes in the current network; the modularity is calculated at this point.(3)For each node *i*, we consider its neighbor j and evaluate the modular gain caused by deleting it from the original community and affiliating it to the other community. We divide it into communities with the largest gain and greater than 0. If the gain of all communities is less than or equal to 0, the node will not carry out community transfer. This process is applied to all nodes repeatedly and sequentially, until there is no further improvement, at which point this step ends. The modular gain is calculated as follows:(4)$$\Delta Q=\left[\frac{\sum_{in}+k_{i,in}}{2m}-{\left(\frac{\sum_{tot}+k_{i}}{2m}\right)}^{2}\right]-\left[\frac{\sum_{in}}{2m}-{\left(\frac{\sum_{tot}}{2m}\right)}^{2}-{\left(\frac{k_{i}}{2m}\right)}^{2}\right],$$
where $\sum_{in}$ is the number of edges in the community *c*, $\sum_{tot}$ is the total degree of the nodes in the community *c*, $k_{i}$ is the degree of node *i*, $k_{i,in}$ is the sum of the number of connections between node *i* and the nodes in community *c*, and *m* is the number of connections in the network.(4)The obtained communities in the previous step are taken as nodes, and a new network is reconstructed. The connection weight between nodes is the sum of all nodes in the original network between the two communities. The nodes have self-circulation, and the weight is the sum of connections of the original nodes in the community. Then, step 3 is repeated for the new network until there are no further gain updates, and the algorithm ends.

The software graph structure can be divided into multiple subgraphs through the Louvain algorithm; therefore, the graph can be represented by $G=\left\{G_{1},G_{2},\ldots ,G_{n}\right\}$. $A_{i}$, $X_{i}$ in $G_{i}\left(A_{i},X_{i}\right)$ respectively represent the adjacency matrix of the edges in the subgraph, and the software defect measurement values of the node.

#### 2.4.2. Learning and Classification of the Node Representation Vector

The explicit feature information of the node and the structural information of the graph network can both be used by the graph neural network to learn the representation vector of the node. It solves the issue that the current software defect prediction model only considers one of the two. The input for the graph neural network model is the data that were obtained following the data processing in Section 2.4.1. A graph neural network’s architecture is shown in Figure 2. The entire framework may be divided into two parts: the node representation vector and the graph convolution process, which learns the representation vector of nodes on top of the graph. The classifier’s design, which is covered in the Section 2, primarily utilizes the multilayer perceptron to classify, and the outcomes of each layer are combined through weights as the final result.
(1)The node representation vector is learned using the graph neural network. Each subgraph undergoes multilayer graph convolution in order for nodes to gain deep semantic information, and each layer’s representation vector is described by the following formula:(5)$$L_{l}=cat\left(H_{0}^{l},\ldots ,H_{i}^{l}\right),l\in \left[0,num\_gcn\right],i=num\_subgraph$$
(6)$$H_{i}^{l+1}=GNN\left(A_{i},H_{i}^{l}\right)$$
where $L_{l}$ represents the representation vector of all nodes of the *L*-th layer, $H_{i}^{l}$ represents the representation vector of all nodes of the *i*-th subgraph after the *L*-th graph convolution, $H_{i}^{0}$ is the initial node information $X_{i}$ of each subgraph, num_subgraph represents the number of all subgraphs of a software, num_gcn represents the number of layers of the convolution layer, the new representation vector $H_{i}^{l+1}$ is obtained by inputting the representation vector $H_{i}^{l}$ of the previous layer of the subgraph and its adjacency matrix $A_{i}$ into the convolution layer of the graph, the cat function concatenates the node representation vectors of all subgraphs into a whole, and GNN is the graph convolution.(2)A classifier is created using graph convolution that learns the representation vector, predicts the output of each layer using MLP, and convolves the output of each layer using a different depth graph. This model chooses to assign a learnable weight to each layer’s output. The representation vector can be utilized more efficiently in this way, and the precise formula is as follows:(7)$$out=\sum_{j=0}^{n}w_{j}\left(MLP_{j}L_{j}\right),n=num\_gcn,$$
where $w_{j}$ is a learnable parameter, the initial value of which is set to $\left(1/n,1/n\right)$, MLP is a multilayer perceptron, whereby each layer representation vector is set with an MLP, L is the representation vector of each layer, num_gcn is the number of graph convolution layers, and out represents the label obtained after the node passes through the model.(3)The pseudocode of the method, which is provided below, presents the process of a thorough Algorithm 1 that demonstrates how each node can learn a representation vector and generate predictions.



**Algorithm 1: Graph neural network learning and prediction**

**Input:**
The graph structure $G=\left\{G_{1},G_{2},\ldots ,G_{n}\right\}$$.\;G_{i}=\left(A_{i},X_{i}\right)$$,\;A_{i}$, and $X_{i}$ represent nodes, the adjacency matrix of the edges, and the software defect measurement element of the nodes, respectively.Output: The prediction result pred of the node1.for *i* in num_layer do2.//num_layer: the number of graph convolutional layers3.//num_subgraph: the number of subgraphs4.*L* = 0;5.for *j* in num_subgraph do6.Put the subgraph $G_{i}=\left(A_{i},X_{i}\right)$ into the graph convolution layer to learn the representation vector of the node;7.end for8.*L* = predicted result of MLP;9.pred + = W × L;10.end for11.return pred


The algorithm mentioned above has two improvements, as can be seen. First, the software source code’s graph structure is initially divided into a number of smaller graphs, which are then used as inputs for graph neural network models. Second, each layer’s prediction results are given some weight.

## 3. Simulation Experiments

### 3.1. Experimental Environment and Datasets

Experiments were performed on the Windows-based operating system, the language used was python [25], and the construction of the graph neural network model was completed through PyTorch [26] and torch-geometric.

The PROMISE dataset [27], a collection of open-source software projects, serves as the dataset in use. Six projects were picked from this dataset, which contains object-oriented measurement elements for all of the dataset’s measurement items. The dataset is described in Table 1.

It can be found that there is a class imbalance problem existing in the data. To improve the dataset, we first used the NearMiss algorithm [28], which reduced the amount of data in the experiment and test. During the experimenting, tenfold cross-validation was used. Each time, 90% of the data were randomly selected for training, 10% of the data were tested, and the results are given using an average of 10 times the data.

### 3.2. Evaluation Measures

To prove the validity of proposed model, the selected evaluation measures such as the accuracy rate, F-measure, and MCC value were used, which were all obtained through the confusion matrix. The confusion matrix [29] is shown in Table 2.

Accuracy refers to the proportion of correct classification to the total number, and the value range is [0, 1]. Higher values indicate better classifier performance. The formula is as follows:(8)$$Acc=\frac{TP+TN}{TP+FN+TN+FP}.$$

The F-measure is the harmonic average of precision rate and recall rate. Precision rate P refers to the proportion of the number of positive samples correctly classified by the classifier to the overall number of positive samples classified by the classifier, and recall rate R refers to the proportion of the number of positive samples correctly classified by the classifier to the number of desired positive samples. The value range is [0, 1], whereby a higher value indicates better classification. The formula is as follows:(9)$$P=\frac{TP}{TP+FP}.$$
(10)$$R=\frac{TP}{TP+FN}.$$
(11)$$\mathrm{F-measure}=\frac{2\times P\times R}{P+R}.$$

MCC is a more appropriate, balanced metric since it takes into account true examples, true-negative examples, false-positive examples, and false-negative examples. The value range is [−1, 1]. A prediction with a value of 1 is considered to be perfect; a prediction with a value of 0 is considered to be only slightly better than a random guess; and a prediction with a value of −1 is considered to be wholly incongruent with the actual result. The equation reads as follows:(12)$$MCC=\frac{TP\times TN-FP\times FN}{\sqrt{\left(TP+FP\right)\left(TP+FN\right)\left(TN+FP\right)\left(TN+FN\right)}}.$$

### 3.3. Experimental Setup

In order to reduce the influence of experimental parameters, some training hyperparameters were set as shown in Table 3.

The parameters used in the traditional method of SVM (support vector machine) were set as default. The network structure of other models are described below. The network structures of the BP neural network comprised four layers. Without using a classifier, GCN was constructed in accordance with the model described in [20], which directly derived the result via a graph convolution operation. The GIN structure was developed in accordance with [23]. The difference is that there was no community division and no final weighted summation. According to the graph neural network framework proposed in this paper, CBGCN (community-based GCN) and CBGIN (community-based GIN) were built. The difference was the graph convolution operation, with the former based on the spectral domain, and the latter based on the spatial domain. The classifier, MLP (multilayer perceptron), and BP neural network were all connected. The specific parameters are shown in Table 4.

### 3.4. Experimental Procedure

This section presents some assumptions and limitations during the experiment, and then describes the specific steps of the experiment. The assumptions in this paper are as follows:(1)We focus on software defect prediction within a project, and the training and testing data are derived from one dataset. For example, when experimenting with ant dataset, the training set is selected and the test set is derived from the remainder of the dataset.(2)During the experiments, a small number of defective classes cause the trained model to favor the non-defective classes. Therefore, class imbalance is applied to the entire dataset before training the model.(3)To better estimate the algorithm performance, a tenfold cross-validation is used.

Under the above assumptions, the validity of the software defect prediction framework proposed in this paper can be verified. The specific experimental procedure is as follows:

Step 1: Using the code analysis tool, the software’s class dependence is extracted, and a CSV file is then generated.

Step 2: The labeled nodes and feature metrics are obtained for the nodes from the PROMISE dataset.

Step 3: NetworkX, a third-party package in python, is used to store the graph structure. Then, the python-louvain package in python is used to divide the graph structure into subgraphs.

Step 4: The NearMiss algorithm is applied to deal with data class imbalance. Then, 90% of the processed dataset is chosen at random for the graph neural network model’s training, and 10% is chosen for its testing.

Step 5: The graph structure from step 3 is used as the input to the graph neural network model. The training set labels are picked in step 4 to train the network parameters.

Step 6: Then, 10% of the data in Step 4 are used for testing, before calculating the performance on various evaluation metrices.

Step 7: The process is repeated 10 times from Step 4 onward.

## 4. Results and Discussion

The spectral domain-based graph convolutional neural network GCN and the spatial domain-based graph convolutional neural network GIN were chosen for studies to show that this model can increase the performance of software defect detection. The models were consequently divided into two groups, the first of which consisted of SVM, BP, GCN, and CBGCN, and the second of which consisted of SVM, BP, GIN, and CBGIN. SVM and BP, the two most fundamental machine learning algorithms, directly use the original feature vector to forecast software problems. Graph convolution is employed by both model frameworks used in the original paper—GCN and GIN—to obtain the characteristics of the graph structure. Two distinct graph convolution techniques were merged by CBGCN and CBGIN to create this model.

### 4.1. Experimental Analysis of Graph Convolutional Neural Network Based on Spectral Domain

In this section, the graph convolution method based on the spectral domain is used as the convolution layer of this model. In order to verify that the graph convolution method can improve the performance of software defect prediction, in this work, in addition to the traditional method, the graph convolutional neural network [20] model was selected as a benchmark model. The results are shown in Table 5.

It can be seen from Table 5 that the proposed model has achieved good experimental results in terms of accuracy, F-measure, and MCC in most datasets. In terms of accuracy, it was 7.6% higher than SVM, 5.8% higher than BP, and 13.6% higher than GCN. The F-measure was 10.7% higher than SVM, 7.8% higher than BP, and 13.1% higher than GCN. The MCC index was 12.5% higher than SVM, 12.7% higher than BP, and 27.4% higher than GCN. The data were analyzed from two aspects:
(1)Comparing CBGCN with SVM and BP, it was found that our model was better than the BP neural network and SVM according to the evaluation of all metrics from other datasets except for the Ant dataset. It was found that, in the Ant dataset, the result of the BP neural network was also lower than that of the SVM. In individual datasets, the parameters of BP need to be specially set to obtain the best performance, and the structure of the CBGCN classifier is the same as the BP neural network. Therefore, individual datasets need to adjust the parameter settings of the network. However, in terms of average, CBGCN was greatly improved; thus, it can be concluded that useful feature vectors can be learned by incorporating the spectral domain-based graph convolution method into this model.(2)Comparing CBGCN with GCN, we found that, except for the Lucene dataset, the experimental results were very similar. Other datasets greatly improved the model, and the average of the evaluation measures was higher; therefore, it can be concluded that the model framework of this paper was more suitable for software defect prediction.

In order to visually observe the performance of the CBGCN algorithm, various evaluation measures were determined as box plots [30]. The y-axis represents the evaluation metric score, while the prediction method is on x-axis in Figure 3. The mean and median values are designated as particular values in the figures to aid in analysis.

Figure 3 shows that the model suggested in this paper had each average evaluation index at its highest point, and that the GCN model’s framework was insufficient for predicting software defects. However, according to the experimental results of CBGCN, the graph convolution method based on the spectral domain could enhance each evaluation index of software defect prediction.

### 4.2. Experimental Analysis of Graph Convolutional Neural Network Based on Spectral Domain

In the experiments in this section, the spatial domain-based graph convolution method is used as the convolution layer of this model. In order to verify that the graph convolution method could still improve the performance of software defect prediction in this work, in addition to the traditional method, the spatial domain-based graph neural network model [23] was selected as a benchmark model. The results are shown in Table 6.

Table 6 shows that, in most datasets, the model proposed in this paper achieved good experimental results in terms of accuracy, F-measure, and MCC. On average, the accuracy was 8.5% higher than SVM, 6.8% higher than BP, and 3.5% higher than GIN. The F-measure was 10.8% higher than SVM, 7.9% higher than BP, and 2.4% higher than GIN. The MCC was 14.6% higher than SVM, 14.7% higher than BP, and 7.0% higher than GIN. The results were analyzed from two aspects:(1)Compared with BP, CBGIN was improved on all datasets. It was still lower than SVM in the Ant project, but higher than CBGCN, demonstrating that the classifier network structure and graph convolution method settings could impact the outcomes. Individual datasets require adjusting the network hyperparameters. Overall, there was a substantial improvement in CBGIN. Thus, it can be inferred that this model may acquire valuable feature vectors by incorporating the spatial domain-based graph convolution method.(2)When CBGIN and GIN were compared, it was discovered that the model was improved across all datasets. We can draw the conclusion that the model presented in this paper is more suited for predicting software defects.

Similarly, in order to visually observe the performance of the CBGIN algorithm, various evaluation metrics were determined as box plots. The y-axis shows the evaluation metric score, while the prediction method is on x-axis in Figure 4. For better analysis, the mean and median in the figure are marked.

Figure 4 shows that the CBGIN was improved with strong performance across all evaluation metrics, suggesting that the GIN model can use the learned representation vector more effectively and that the enhanced GIN model is more suited for software fault prediction.

The experimental results demonstrated that both the GCN and the GIN graph convolution methods can produce beneficial representation vectors to enhance the accuracy of software defect prediction. This model was improved compared to GIN and GCN, showing that the model suggested in this research can increase the accuracy of software defect prediction.

## 5. Conclusions and Future Work

In this paper, we mapped the software to the graph structure and simplified the software graph with the community structure according to the complex network theories. Furthermore, we used the convolutional layer in the graph neural network to obtain the graph information of the software. In this way, the software was regarded in its entirety, and independent classes were linked through class dependencies for software defect prediction. The graph convolution layer selected the graph convolution method as GCN and GIN for experiments and used the PROMISE dataset for verification. The experimental results show that the graph neural network could obtain better representation vectors of nodes, thereby improving the performance of software defect prediction. 

This research highlights the importance of a software defect prediction framework based on multiple factors, by modeling the software into a more complex network, which considers the connections between the software modules and the attributes of modules. The following suggestions for future work can be derived from this experiment:(1)A more complex network can be constructed, for example, considering developer information, and semantic information of software code can be incorporated into the network.(2)For the improvement of the graph neural network, it can be combined with a community discovery algorithm.(3)The experiments in this paper considered within-project software defect prediction; thus, in the future, cross-project software defect prediction can be considered.

## Figures and Tables

**Figure 1 entropy-24-01373-f001:**
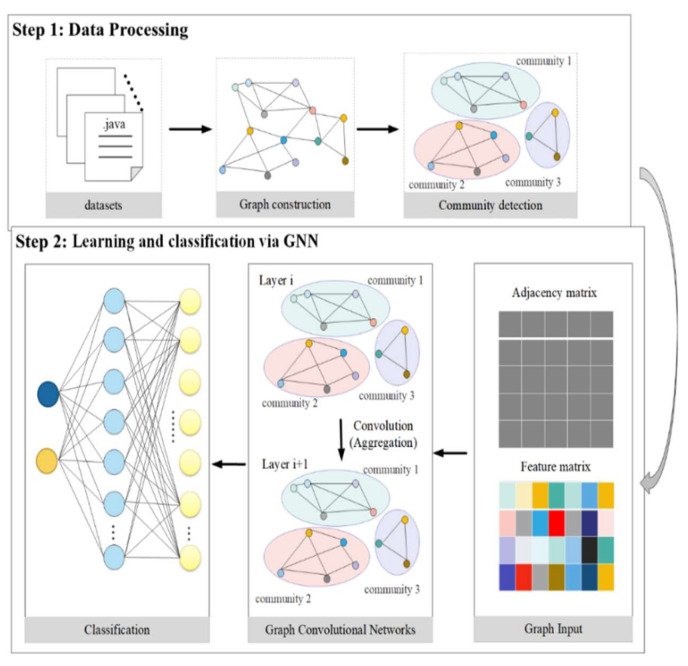
The general framework of this model.

**Figure 2 entropy-24-01373-f002:**
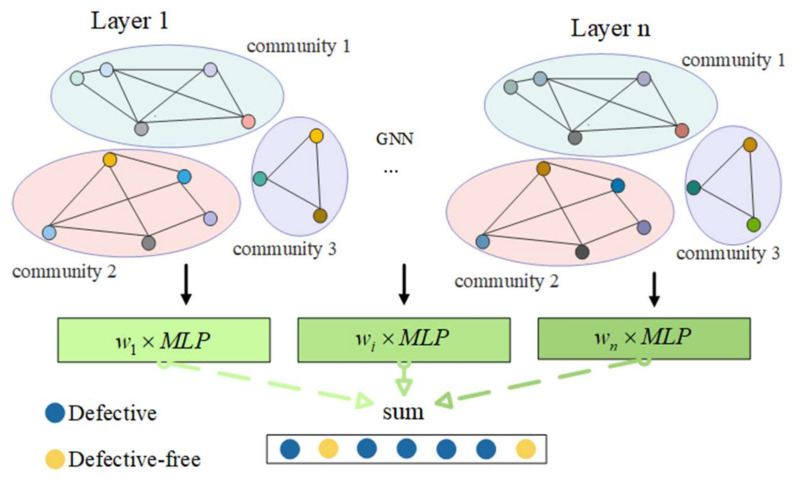
The framework of graph neural network.

**Figure 3 entropy-24-01373-f003:**
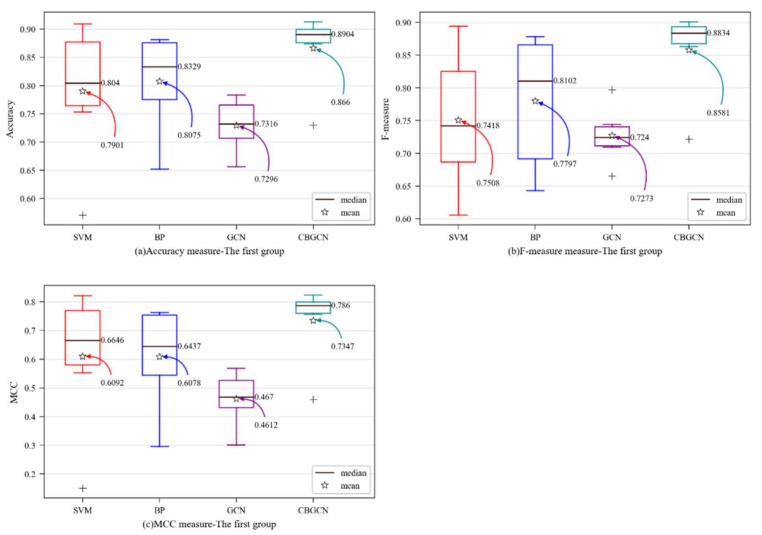
Dataset-wise boxplots of the first group: (**a**) accuracy; (**b**) F-measure; (**c**) MCC.

**Figure 4 entropy-24-01373-f004:**
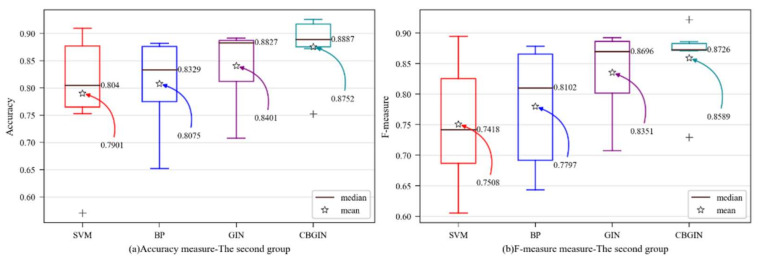
Dataset-wise boxplots of the second group: (**a**) accuracy; (**b**) F-measure; (**c**) MCC.

**Table 1 entropy-24-01373-t001:** Description of the dataset collected from PROMISE.

Datasets	Version	Number of Features	Number of Nodes	Number of Edges	Number of Defects	Defect Rate
Ant	1.7.0	20	745	3961	166	0.2228
Camel	1.6.0	20	965	4215	188	0.1948
Lucene	2.4	20	340	1559	203	0.5970
Synapse	1.2	20	256	1162	86	0.3359
Velocity	1.6.1	20	229	1292	78	0.3406
Ivy	2	20	352	2063	40	0.1136

**Table 2 entropy-24-01373-t002:** Confusion matrix.

Actual Label	Predicted Label	
	Defective	Defective-Free
Defective	TP (true positive)	FN (false negative)
Defective-free	FP (false positive)	TN (true negative)

**Table 3 entropy-24-01373-t003:** Training parameters.

Parameter	Setting
Number of iterations	10,000
Learning rate	0.0001
Weight_decay	5 × 10^−4^
Loss function	CrossEntropyLoss
Optimizer	Adam
Activation function	Relu

**Table 4 entropy-24-01373-t004:** Network parameters.

Algorithm	Number of Graph Convolution Layers	Change of Vector Dimension	Number of Layers of Classifier	Change of Vector Dimension
BP	0	None	4	20, 10, 10, 2
GCN	2	20, 16, 2	0	None
CBGCN	4	20, 20, 20, 20, 20	4	20, 10, 10, 2
GIN	4	20, 20, 20, 20, 20	2	20, 2
CBGIN	4	20, 20, 20, 20, 20	2	20, 2

**Table 5 entropy-24-01373-t005:** Comparison between CBGCN and other prediction methods.

Dataset	Evaluation Measures	SVM	BP	GCN	CBGCN
Ant	Accuracy	0.9091	0.8794	0.7000	0.8735
	F-measure	0.8942	0.8656	0.7178	0.8629
	MCC	0.8208	0.7614	0.4218	0.7553
Camel	Accuracy	0.8081	0.8658	0.7263	0.8974
	F-measure	0.7546	0.8655	0.7440	0.8965
	MCC	0.6682	0.7309	0.4589	0.7985
Lucene	Accuracy	0.5704	0.6519	0.7370	0.7296
	F-measure	0.6054	0.6431	0.7301	0.7214
	MCC	0.1496	0.2955	0.4751	0.4598
Synapse	Accuracy	0.7529	0.7667	0.7833	0.8833
	F-measure	0.6727	0.7550	0.7971	0.8849
	MCC	0.5526	0.5564	0.5681	0.7748
Velocity	Accuracy	0.8000	0.8812	0.6562	0.9000
	F-measure	0.7290	0.8784	0.6651	0.9007
	MCC	0.6610	0.7631	0.3003	0.7973
Ivy	Accuracy	0.9000	0.8000	0.7750	0.9125
	F-measure	0.8489	0.6705	0.7094	0.8820
	MCC	0.8030	0.5393	0.5429	0.8224

**Table 6 entropy-24-01373-t006:** Comparison between CBGIN and other prediction methods.

Dataset	Evaluation Measures	SVM	BP	GIN	CBGIN
Ant	Accuracy	0.9091	0.8794	0.8912	0.8853
	F-measure	0.8942	0.8656	0.8833	0.8712
	MCC	0.8208	0.7614	0.7869	0.7790
Camel	Accuracy	0.8081	0.8658	0.8842	0.8921
	F-measure	0.7546	0.8655	0.8877	0.8858
	MCC	0.6682	0.7309	0.7754	0.7954
Lucene	Accuracy	0.5704	0.6519	0.7074	0.7519
	F-measure	0.6054	0.6431	0.7075	0.7295
	MCC	0.1496	0.2955	0.4124	0.5121
Synapse	Accuracy	0.7529	0.7667	0.7889	0.8722
	F-measure	0.6727	0.7550	0.7838	0.8715
	MCC	0.5526	0.5564	0.5890	0.7554
Velocity	Accuracy	0.8000	0.8812	0.8812	0.9250
	F-measure	0.7290	0.8784	0.8925	0.9218
	MCC	0.6610	0.7631	0.7640	0.8514
Ivy	Accuracy	0.9000	0.8000	0.8875	0.9250
	F-measure	0.8489	0.6705	0.8559	0.8737
	MCC	0.8030	0.5393	0.7821	0.8378

## Data Availability

Not applicable.

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
