# Peer review of "Research of Software Defect Prediction Model Based on Complex Network and Graph Neural Network"

_entropy, 2022, doi:10.3390/e24101373_

Round 1
Reviewer 1 Report
This is an interesting work presenting a GNN solution for software defect predicition. Some comments:
- The work is really difficult to follow: the meaning of many english sentences are not clear and the description of the methods are very confusing.
- To have access to the code would be very important to better assess the methods used.
- Highlight all assumptions and limitations of your work.
- Conclusions should provide some lessons learnt.
Reviewer 2 Report
The article entitled “Research of Software Defect Prediction Model Based on Complex Network and Graph Neural Network” is well-written and, from my point of view, would be of interest for the readers of Entropy. In spite of these and before its publication, I consider that authors should perform the following changes in order to improve the quality of the manuscript. The changes suggested are as follows:
In lines 69-70 it is said: “mapped the software as a dependent class-based network, using different graph em-69 bedding techniques to obtain representation vectors of the graph”. From my point of view, it would be difficult for readers to understand it. Please, explain more in-depth.
At the end of the introduction section please also explain how the rest of the paper is arranged.
Line 104 it speaks about “Seven Bridges of Konigsberg of network science” please introduce a reference about it.
In section 2.4. Software defect prediction model based on complex network and graph neural network, the content of Figure 1 should be explained more in-depth and, if possible, Figure 1 should be divided in two.
In lines from 270 to 272 it is said “Experiments are done on the windows-based operating system, the language used is 270 python, and the construction of the graph neural network model is completed through PyTorch and torch-geometric”. At least one reference to Python and another to PyTorch is required.
Table 1: please review the header of this table and leave the text in a way easy to read.
Table 3: please express the weight decay with the right scientific notation.
Reviewer 3 Report
The manuscript “Research of Software Defect Prediction Model Based on Complex Network and Graph Neural Network” proposed a software defect prediction framework based on complex network science.
This manuscript is confusing and I had problems understanding the results and the meaning of this research. I find many flaws, obscure parts, strange Refs format, and strange sentences. I suggest a careful revision of the language.
Another major problem is the claimed relationship between the methods presented in this manuscript and the software defection problem. It is not clear the utility of the method presented in this paper and the real problem under study. On the other hand, I am not convinced that the methods presented here may be useful to detect software defections.
OTHERS
- Which is the Software used to process and analyse networks? For example, it is important to know what is the software adopted to compute modularity Q.
- The manuscript presents many acronyms scattered throughout the paper. I suggest adding a table of acronyms to make easier the reading.
-R101: This sentence seems non-sense: “The complex network [10] is a method for analyzing complex systems.”. Further, the Ref 10 in this sentence is a very specific manuscript with a specific complex network application, not a research focusing on complex network science in a general way as suggested by the sentence.
- R103: The sentence: “Complex networks have been developed from the original Seven Bridges of Konigsberg of network science.” Should be “Complex networks have been developed from the original Seven Bridges of Konigsberg PROBLEM of network science.”
- R121: “Modular Q was first presented by Newman [12] in 2004” should be “Modularity Q was first presented by Newman and Girvan [12] in 2004”
- R124: As above, the inline Refs should consider whether the cited manuscript has more than one author. The sentence: “…proposed by Blondel is widely used because of its ability to…” should be “proposed by Blondel et al. is widely used because of its ability to…”.
- R139: ‘Dr. Franco’ ??!? The second name of the first author of the Ref is Scarselli.
Why Dr. Franco ?
- R191: “First define a modular scalar Q…”. ‘Modular scalar Q is wrong. The name of this measure is modularity Q, and it is a measure, not a scalar.
- R266: : What is this part: “This section may be divided by subheadings. It should provide a concise and precise description of the experimental results, their interpretation, as well as the experimental conclusions that can be drawn.”
- R347: What is this part: “Authors should discuss the results and how they can be interpreted from the perspective of previous studies and of the working hypotheses. The findings and their implications should be discussed in the broadest context possible. Future research directions may also be highlighted.”
- R472: What is this part: “References must be numbered in order of appearance in the text (including citations in tables and legends) and listed individually at the end of the manuscript. We recommend preparing the references with a bibliography software package, such as EndNote, ReferenceManager or Zotero to avoid typing mistakes and duplicated references. Include the digital object ……”
Round 2
Reviewer 1 Report
Ok to publish
Reviewer 3 Report
I willingly refer the choice to accept this manuscript to the editor, avoiding further revision of the manuscript.